# A physics-informed machine learning (PIML) framework for projecting 21st-century permafrost extent in Northeast China

Shuai Huang<sup>1, 2, 3</sup>, Xiangbing Kong<sup>3</sup>, Xue Yang<sup>1, 4, 5</sup>, Xiaoying Jin<sup>1, 2</sup>, Shanzhen Li<sup>4</sup>, Lin Yang<sup>1, 4, 6</sup>, Yaodan Zhang<sup>1, 4, 7</sup>, Kai Gao<sup>8, 9</sup>, Hongwei Wang<sup>8</sup>, Xiaoying Li<sup>8</sup>, Ruixia He<sup>8</sup>, Lanzhi Lü<sup>8</sup>, Guodong Cheng<sup>8</sup> and Huijun Jin<sup>1, 2, 4, 5</sup>

- <sup>1</sup>School of Ecology, Northeast Forestry University, Harbin 150080, China;
- <sup>2</sup>Key Laboratory of Sustainable Forest Ecosystem Management (Ministry of Education), College of Forestry, Northeast Forestry University, Harbin 150080, China;
- <sup>3</sup>Département de mathématiques, informatique et génie, Université du Québec à Rimouski, Rimouski G5L 3A1, Canada;
- 4School of Civil Engineering and Transportation, Permafrost Institute, and Ministry of Education Northeast-China Station of Permafrost Geo-environmental Systems (Erguna, Inner Mongolia), Northeast Forestry University, Harbin 150040, China;
   5Ministry of Natural Resources Field Observation and Research Station of Permafrost and Cold Regions Environment in the Da Xing'anling Mountains at Mo'he, Northeast China, Natural Resources Survey Institute of Heilongjiang Province, Harbin 150036, China;
- 15 <sup>6</sup>College of Agriculture, Chifeng University, Chifeng 010019, China;
  - <sup>7</sup>College of Civil Engineering and Architecture, Quzhou University, Quzhou 324000, China;
  - <sup>8</sup>State Key Laboratory of Cryospheric Science and Frozen Soil Engineering, Northwest Institute of Eco-Environment and Resources, Chinese Academy of Sciences, Lanzhou 730000, China, and;
  - <sup>9</sup>University of Chinese Academy of Sciences, Beijing, 100049, China
- 20 Correspondence to: Huijun Jin (hjjin@nefu.edu.cn) and Xiangbing Kong (Xiangbing kong@uqar.ca)

**Abstract.** The degradation of marginal permafrost is a sensitive indicator of climate change, with far-reaching implications on regional ecosystems, hydrology, and infrastructure. Located near the southern limit of latitudinal permafrost (SLLP) in Eastern Asia, Northeast China has experienced pronounced permafrost retreat and persistent ground warming in recent decades. This study develops a physics-informed machine learning (PIML) framework that integrates the Temperature at the Top of Permafrost (TTOP) model, observed changes in land use and land cover (LULC), and climate projections from the Coupled Model Intercomparison Project 6 (CMIP6) to improve the understanding and prediction of permafrost dynamics in the region. Results indicate that, under the SSP5-8.5 scenario, permafrost extent may decline by more than 90% by the end of the 21st century, primarily driven by a sharp reduction in the air freezing index (AFI), especially in high-latitude and highelevation zones. Land use and cover changes (LUCC), particularly urban expansion and deforestation, further exacerbate ground thermal disturbances. Spatially, mountainous forested areas, such as the Da Xing'anling Mountains, exhibit relatively greater resilience to warming due to dense vegetation and complex topography that help buffer surface energy fluxes. Feature attribution analysis identifies surface temperature, snow cover duration, and vegetation as dominant drivers of permafrost stability, while Uniform Manifold Approximation and Projection (UMAP) clustering reveals distinct degradation trajectories across different land cover types. This study highlights the complex interplay of climatic and anthropogenic factors in permafrost evolution and demonstrates the utility of integrating physical modelling with machine learning to support ecological conservation and infrastructure risk management in cold regions environment.

#### 1 Introduction

60

Permafrost, a crucial component of the Earth's cryosphere, underlies approximately 11% of the currently exposed land surface and is highly sensitive to climate warming (Obu, 2021). Recent decades have seen disproportionately rapid climate warming in northern high-latitude regions, a phenomenon known as Arctic amplification (You et al., 2021). Correspondingly, Arctic and subarctic permafrost has experienced unprecedented warming; for instance, a synthesis of observational records indicates that average global permafrost temperature have increased by approximately  $0.29 \pm 0.12$  °C over the past decade (Biskaborn et al., 2019). As a result, previously stable permafrost terrains are beginning to thaw, underscoring their acute vulnerability to ongoing climate change (Tang et al., 2024).

The degradation of permafrost has cascading impacts on ecosystems, hydrology, geomorphology, infrastructure, and biogeochemical cycles. By decoupling carbon and nitrogen dynamics, thawing permafrost accelerates greenhouse gas emissions, weakening the carbon sink capacity of northern ecosystems and potentially tipping them toward becoming net carbon sources (Mackelprang et al., 2011; Koven et al., 2015; Chen et al., 2018; Liu et al., 2022). The thaw of ice-rich permafrost can also induce ground subsidence and surface deformation, transforming stable terrain into wetlands and peat bogs, disrupting hydraulic connectivity, and altering ecosystem structure and function (Olefeldt et al., 2016, 2021; Walvoord and Kurylyk 2016; Jin et al., 2021). Moreover, thawing permafrost may release viable microorganisms, ancient viruses, radon, and sequestered contaminants, posing emerging threats to public health and environmental security (Miner et al., 2021; Zhang et al., 2024a).

Northeast China, once characterized by relatively delayed and dwindled warming, has experienced a marked climatic shift since the mid-20th century, signaling a distinct regional climate shift (Jin et al., 2000, 2007, 2016, 2019, 2020, 2025; Li et al., 2020). Following the end of the global warming hiatus (Chang et al., 2024), permafrost extent in the region has declined significantly, shrinking by approximately 12%, from 2.39×10<sup>5</sup> km<sup>2</sup> in the late 1990s to 2.10×10<sup>5</sup> km<sup>2</sup> in the late 2010s (Wang et al., 2024a). Concurrently, the southern limit of latitudinal permafrost (SLLP) has retreated northward by 50-120 km, and permafrost thawed or detached, forming taliks of various types in some southern areas (Jin et al., 2007, 2025; Li et al., 2022a). As a sensitive indicator of climate change near the Eastern Asian cryospheric fringe, the permafrost in Northeast China is preserved in a delicately balanced state, making it particularly susceptible to early degradation under warming scenarios (Jin et al., 2007). These changes not only offer a preview of future transformations in more northerly continuous permafrost zones but also serve as forerunners of large-scale cryospheric transitions. The rapid thaw observed in Northeast China (Li et al., 2022b; Huang et al., 2023; Wang et al., 2023) foreshadows broader permafrost responses under sustained warming. Moreover, the degradation of this marginal permafrost threatens hemiboreal ecosystems and cryospheric hydrological regimes, potentially triggering disproportionate ecological shifts—such as boreal forest retreat, wetland loss, and other biogeographic reorganizations (Baltzer et al., 2014; Li et al., 2021). Consequently, modeling and predicting permafrost changes in this region holds not only regional but also global significance for understanding climate-cryosphere feedbacks.


Accurate prediction of permafrost dynamics requires robust modeling of permafrost-climate interactions. Over the past five decades, three primary classes of permafrost models have emerged: empirical, equilibrium, and physical (numerical) models (Riseborough et al., 2008). Empirical models rely on statistical relationships between permafrost conditions and environmental variables, such as air temperature, vegetation cover, and elevation, and include approaches such as the basal temperature of snow cover (BTS) model (e.g., Lewkowicz and Ednie, 2004), Gaussian model (e.g., Cheng, 1984; Li, 2024), and frost indices (e.g., Nelson and Outcalt, 1987). Although these models are computationally efficient and well-suited for regional mapping, they are constrained by data availability and exhibit limited generalizability under non-stationary climate conditions.

Equilibrium models take a semi-analytical approach, such as the Kudryavtsev model (Kudryavtsev et al., 1974), the Temperature at the Top of Permafrost (TTOP) model (Smith and Riseborough, 2002), and the Stefan model (Shiklomanov and Nelson, 2002), assume a steady-state ground thermal regime in response to long-term climate averages (Riseborough et al., 2008). They have been widely applied in various permafrost regions, such as Northeast China (Li et al., 2022b; Huang et al., 2023), High-mountain Asia (Kim et al., 2024), and the Northern Hemisphere (Obu et al., 2019) but cannot capture transient dynamics or short-term variability in permafrost hydrothermal dynamics, limiting their application under rapid climate change.

Physical (numerical) models simulate heat transfer through soil using the Fourier heat conduction equation with phase change, often incorporating snow insulation, latent heat, and soil moisture dynamics (Riseborough et al., 2008; de Bruin et al., 2023; Tubini et al., 2023). Examples range from site-specific ground thermal models to large-scale land surface models within climate models (Qin et al., 2017; Sedaghatkish et al., 2024). While highly process-based and suitable for transient simulations, these models mandate extensive and detailed input data (e.g., soil properties, vegetation cover, and geothermal flux) and are computationally intensive. Moreover, uncertainties in parameterization and incomplete representation of subgrid heterogeneity can result in substantial variability in model projections (Groenke et al., 2023; Wang et al., 2024b).

avenues for permafrost modeling (Luo et al., 2024). ML approaches can learn complex, non-linear relationships from satellite observations, reanalysis datasets, and in situ measurements. Recent studies have employed ML to map permafrost distributions and predict active layer thickness or ground temperatures with high spatial resolution (Ran et al., 2021; Thaler et al., 2023; Chance et al., 2024; Zhang et al., 2024b; Zou et al., 2025). ML can also emulate computationally expensive physical models once adequately trained (Luo et al., 2024). However, challenges remain, especially the scarcity of high-quality observational data in remote permafrost areas (Fatolahzadeh Gheysari and Maghoul, 2024; Chang et al., 2024).

The emergence of machine learning (ML) techniques and the proliferation of large environmental datasets have opened new

Additionally, a key limitation of conventional ML approaches lies in their lack of physical interpretability and inability to explicitly incorporate governing processes. This becomes especially problematic when modeling permafrost thaw, which is often governed by threshold behaviors and nonlinear feedbacks (e.g., retrogressive thaw slumps) that are difficult to capture from limited training data alone (Painter et al., 2013; Chang et al., 2024). To address these issues, hybrid approaches such as physics-informed machine learning (PIML) have gained traction (Karniadakis et al., 2021; Pilyugina et al., 2023). These








approaches combine physical laws with data-driven ML, either by embedding known physical constraints into ML algorithms or by using ML to calibrate and enhance physical models, thereby improving interpretability and generalizability. Despite these advances, one critical gap persists: the limited incorporation of land use and cover changes (LUCC). While climate is the primary driver of permafrost degradation, human activities, such as deforestation, agriculture, urbanization, and infrastructure expansion, can significantly alter ground thermal conditions (Wang et al., 2022; Jin et al., 2024). These processes are particularly impactful in regions such as Northeast China, where centuries of land transformation have drastically reshaped the natural landscape. Since the mid-19th century, following national development policies and land exploitation, the region has experienced widespread destruction of its original boreal forests, leading to sharp declines in natural insulation and hydrological stability. However, many current permafrost models continue to assume static land cover conditions (e.g., Hu et al., 2022; Zhang et al., 2022; Peng et al., 2023), thus failing to capture the complexity and magnitude of degradation in dynamically evolving landscapes. This modeling simplification has been shown to significantly underestimate of ground temperature increases and active layer deepening (Serban et al., 2021; Peplau et al., 2023). For example, long-term monitoring by remote sensing in the Hola Basin, Northeast China revealed rapid forest loss and anthropogenic land expansion from 1973-2019, which coincided with accelerated permafrost degradation and the onset of thermokarst hazards (Serban et al., 2021). Similarly, field experiments in subarctic Canada demonstrated that agricultural conversion led to soil warming exceeding regional climate trends, highlighting the direct role of LUCC in amplifying ground heat flux and organic matter decomposition (Peplau et al., 2023). Modelling studies based on Earth system simulations and paleoclimate experiments further confirm that LUCC exerts both biogeophysical and biogeochemical influences on permafrost, through altered albedo, evapotranspiration, and carbon cycling (Peng et al., 2020, 2025). Ignoring these drivers in predictive frameworks not only limits the accuracy of permafrost extent projections but also undermines assessments of permafrost carbon feedbacks, hydrological change, and engineering or ecological risk under future scenarios (Ward Jones et al., 2024; Jin et al., 2025). This discrepancy can lead to fundamental errors in downstream estimations of carbon emissions, water resource shifts, infrastructure stability, and environmental adaptation strategies. Therefore, incorporating dynamic LUCC trajectories into permafrost modelling is not only scientifically necessary for capturing coupled land-climate feedback, but also practically critical for delivering reliable projections in regions undergoing rapid land transformation. In response to these challenges, this study develops a PIML-based permafrost modelling framework that integrates the TTOP model with ML to enforce physical constraints. The framework incorporates LUCC projections from the Patch-generating Land Use Simulation (PLUS) model and uses downscaled CMIP6 outputs in combination with meteorological station data to drive permafrost forecasts. By simulating ground temperature evolution and permafrost extent under future climate and land use scenarios, this study offers a physically grounded, data-enhanced approach to understanding permafrost dynamics in Northeast China. The findings provide a valuable basis for ecological conservation, infrastructure planning, and climate adaptation in cold regions, while contributing broader insights into permafrost-climate feedbacks on a global scale.

## 2 Methods





## 2.1 Meteorological data downscaling

Daily observational meteorological data from 225 weather stations in Northeast China covering the period 1961–2020) were obtained from the National Meteorological Information Centre of China (NMICC) (Fig. S1). These station records served as baseline climate data for downscaling. To properly represent a range of climate projections for the study area, 14 global climate models (GCMs) were selected from the CMIP6 ensemble: ACCESS-ESM1-5, BCC-CSM2-MR, CanESM5, CESM2-WACCM, CMCC-ESM2, CNRM-CM6-1, CNRM-ESM2-1, INM-CM4-8, INM-CM5-0, IPSL-CM6A-LR, MIROC6, MRI-ESM2-0, NorESM2-LM, and NorESM2-MM. Key characteristics of each model, including modeling center, land surface scheme, and native resolution, are summarized in Table S1 for comparison. A statistical delta downscaling method is applied to bias-correct the GCM outputs using the NMICC station baseline as reference data (Navarro Racines et al., 2020). The corrected future air temperature is calculated as:

$$T_{\text{fut}}^{\text{bc}} = T_{\text{fut}}^{\text{mod}} + \left(T_{\text{hist}}^{\text{obs}} - T_{\text{hist}}^{\text{mod}}\right),\tag{1}$$

where  $T_{\text{fut}}^{\text{mod}}$  is the raw model-projected future temperature;  $T_{\text{hist}}^{\text{obs}}$ , the observed historical mean temperature from meteorological stations, and;  $T_{\text{hist}}^{\text{mod}}$ , the model-simulated historical temperature.

In this approach, long-term monthly air temperature anomalies (future changes relative to a historical reference period) were first computed from each CMIP6 model for variables such as daily mean air temperature. These model-derived anomalies were then superimposed onto the historically observed station data, generating localized climate projections. This procedure preserves the observed temporal patterns and extremes at each station while incorporating model-projected climate change signals. The resulting delta-downscaled dataset comprises daily air temperature series at each station, reflecting the climate trends projected by each GCM. This downscaled meteorological dataset, which captures local climate variability, is used in all subsequent analyses within the permafrost modelling workflow (Fig. S2).

## 2.2 Land use projection using PLUS model

Land use changes at an 1-km resolution were projected using the PLUS model (Liang et al., 2018, 2021). As inputs, the fine-resolution China Land Cover Dataset (CLCD) for the years 2000 and 2020 were utilized as the baseline LULC maps. The CLCD provides 30-m land cover classifications for all of China (Yang and Huang, 2021), which were resampled to 1-km resolution to support regional-scale modelling. The two time slices (2000 and 2020) were used to calibrate LUCC dynamics over the 20-year period. The PLUS model is a raster-based cellular automata framework for patch-level LUCC, extending the conventional Cellular Automata (CA)–Markov approach with the Land Expansion Analysis Strategy (LEAS) module and self-adaptive patch generation mechanism (Cellular Automata for Refined Simulation (CARS) module) (Liang et al., 2018). We first derived the land transition demand by applying a Markov chain analysis to the LUCC during 2000–2020, which yielded transition probabilities and the expected area of each land cover type in future years. In other words, the Markov






chain embedded in PLUS was used to predict the quantity of land use demand for a future target year based on observed class changes. Next, the spatial allocation of these changes was handled by the CA-based patch generation module of PLUS, which converts the demand into realistic land use patterns by accounting for neighbourhood effects and drivers of change.

A set of 16 driving factors was supplied to the PLUS model to influence the probability of land conversion in each 1-km grid cell. These variables include natural factors (e.g. elevation, slope angle, soil types, and soil erosion types) and climatic factors (e.g., mean annual precipitation and air temperature from 2000 to 2020), and socio-economic and accessibility indicators (e.g., distances to the nearest expressway, primary, secondary, and tertiary roads, railway, water bodies, existing settlements, and administrative centres, as well as gridded GDP and population density) (Table S2). Each driving factor was prepared as a gridded layer at an 1-km resolution. The LEAS module within the PLUS model employed RF algorithms to quantify relationships between historical LUCC and its driving factors, producing a probability surface of future development for each land-cover category (Fig. S3). The model was calibrated by simulating land use in 2020 using 2000 as the baseline and validated against the actual 2020 map. The Kappa coefficient and overall accuracy were found to be high (both >0.85 in calibration tests), confirming the reliability of the PLUS model in reproducing the observed land changes. After validation, a future land use scenario was generated under a "natural development" assumption (continuation of 2000–2020 trends, without new policy interventions). All model parameters and outputs, including the driver importance and scenario assumptions, are documented in Table S3 for transparency.

#### 2.3 TTOP model

The TTOP model (Smith and Riseborough, 2002) was applied to estimate mean annual ground temperatures (MAGTs) based on near-surface air temperature inputs. The model relates ground thermal conditions to the air freezing and thawing degree-days (FDD<sub>a</sub> and TDD<sub>a</sub>), adjusted by empirical n-factors and thermal offsets as follows:

$$\text{MAGT} = \begin{cases} \frac{1}{\tau} \left( n_f \cdot \text{FDD}_{\mathbf{a}} + n_t \cdot r_k \cdot \text{TDD}_{\mathbf{a}} \right) & \text{for } n_f \cdot \text{FDD}_{\mathbf{a}} + n_t \cdot r_k \cdot \text{TDD}_{\mathbf{a}} \le 0 \\ \frac{1}{\tau} \left( \frac{1}{r_k} n_f \cdot \text{FDD}_{\mathbf{a}} + n_t \cdot \text{TDD}_{\mathbf{a}} \right) & \text{for } n_f \cdot \text{FDD}_{\mathbf{a}} + n_t \cdot r_k \cdot \text{TDD}_{\mathbf{a}} > 0 \end{cases}, \tag{2}$$

where  $\tau$  is the number of days in a year (typically 365), and; FDD<sub>a</sub> and TDD<sub>a</sub> are the freezing and thawing degree-days of air temperature, respectively. The empirical parameters  $n_f$ ,  $n_t$  and  $r_k$  serve to account for snow insulation, surface energy exchange, and the ratio of soil thermal conductivity in frozen to thawed states, respectively.

The model parameters  $n_f$ ,  $n_t$  and  $r_k$  were assigned based on LULC classifications. Each LULC type was associated with a predefined parameter range reflecting differences in surface insulation (e.g. snow cover), vegetation structure, and soil properties, as summarized in Table S4.

The simulated MAGT was derived from 200 realizations, each calculated by randomly sampling the parameters of and nt within a  $\pm 20\%$  perturbation range around their LULC-specific values. From these simulations, we calculated both the





average MAGT and the Permafrost Zonation Index (PZI), defined as the probability that MAGT is below 0 °C across the 200 samples.

To classify the spatial-continuity-based zones of permafrost, we adopted a categorization scheme based on the PZI thresholds originally proposed by Westermann et al. (2015). This approach has been widely adopted and validated in subsequent permafrost modelling, including Obu et al. (2019) and Kim et al. (2024), particularly in applications of TTOP-based models. Specifically, we adopted the classical classification scheme based on areal continuity of permafrost (Brown et al., 1997) and employed the PZI as a quantitative analogue to represent it. Based on PZI values, permafrost was classified into four categories: continuous permafrost (PZI  $\in$  [0.9, 1.0]), discontinuous permafrost (PZI  $\in$  [0.5, 0.9)), sporadic permafrost (PZI  $\in$  [0.1, 0.5)), and isolated patches permafrost (PZI  $\in$  [0.005, 0.1)). This approach enables a probabilistic representation of areal continuity of permafrost, allowing for a quantitative characterization of its distribution status Westermann et al. (2015).

To establish a direct relationship between PZI and MAGT, we performed regression analysis on the simulated results (Fig. S4) and determined threshold MAGT values corresponding to the different permafrost continuity classes. It is noteworthy that under the probabilistic framework of the PZI classification, certain regions with simulated MAGT slightly above 0 °C may still exhibit permafrost-present (i.e., PZI > 0), indicating the potential presence of (buried/detached) marginal permafrost. This arises from the sub-grid heterogeneity incorporated in the ensemble simulations, where variability in snow insulation, vegetation cover, and soil thermal properties allows some realizations to yield MAGT below 0 °C even when the ensemble-averaged MAGT at that grid cell is above 0 °C. Specifically, thick snow cover can substantially weakens the ground cooling in winter, while variations in soil moisture and organic matter content modulate thermal conductivity, leading to localized cold zones that preserve permafrost beneath a relatively warm surface (e.g., Jin et al., 2008; He et al., 2021). Consequently, the PZI-based approach offers a more physically consistent representation of permafrost continuity, especially near the SLLP.

# 2.4 PIML framework

Accurate mapping of ground thermal regimes demands a method that is simultaneously physically plausible, data-adaptive, and computationally tractable. Traditional empirical models, such as the TTOP, incorporates first-principles energy balance constraints; however, their simplified parameterizations limit their accuracy in heterogeneous landscapes. Classical data-driven ML models excel at capturing nonlinear relationships among multiple variables, but their black-box nature prevents a faithful representation of key physical processes, such as heat transfer and phase changes, thereby limiting their ability to generalize beyond the training domain.

To improve the spatial prediction of thermal state of permafrost (TSP), we developed a PIML framework that integrates process-based TTOP model outputs with data-driven modelling. For each 1-km grid cell, we extracted a set of environmental driving variables related to ground thermal conditions, including topographic features (slope angle and slope aspect),





geographic location (latitude, longitude, and elevation), soil properties (bulk density, texture fractions, and organic carbon content), and surface and climatic characteristics (e.g., snow cover duration, land surface temperature, vegetation index, and LULC) (Table S5).

We then constructed a training dataset using the TTOP-estimated MAGT as the target variable, allowing the supervised learning models to capture the relationships between environmental predictors and ground temperature. The dataset was split into a training set (70%) and a testing set (30%) using stratified random sampling to preserve the distribution of key environmental features across the study area (Kuhn et al., 2013). Six commonly used supervised learning algorithms were tested, including DT Regressor, MLP, CatBoost, RF, SVM, and XGBoost. The trained PIML models were subsequently evaluated on the test set and used for spatial mapping of MAGT across the study area.

# 2.5 Variable importance and interpretability

To evaluate the contribution of each environmental predictor to the output of ML models, we employed two model-agnostic interpretability techniques: permutation feature importance and SHAP. Permutation importance quantifies how much a model relies on a given feature by measuring changes in prediction errors when the values of this feature are randomly permuted (Breiman, 2001). In our analysis, model performance was evaluated using RMSE. The importance Ij of a feature xj is defined as:

$$I_j = \text{RMSE}_{\text{permuted(j)}} - \text{RMSE}_{\text{baseline}}, \tag{3}$$

where RMSE<sub>baseline</sub> is the RMSE of the model on the original validation dataset, and RMSE<sub>permuted(j)</sub> is the RMSE after permuting feature  $x_j$ . A greater increase in RMSE indicates more predictive information relevant to the target variable (MAGT).

To further interpret how individual features contribute to specific predictions, we applied SHAP based on cooperative game theory (Lundberg and Lee, 2017). SHAP expresses the model output for a given input x as the sum of the contributions of all features:

$$f(x) = \phi_0 + \sum_{j=1}^{M} \phi_j$$
, (4)

where f(x) is the predicted value,  $\phi_0$  is the base value (i.e. the mean model output over the training data), and  $\phi_j$  is the SHAP value representing the contribution of feature  $x_j$  to the prediction. SHAP values are locally accurate and consistent, making them suitable for interpreting complex models, such as ensemble trees and neural networks. By computing SHAP values for all training samples, we obtained a matrix  $\Phi \in \mathbb{R}^{N \times M}$ , where each element represents the contribution of a given feature to a particular prediction.

To visualize the structure in these high-dimensional SHAP value profiles, we applied UMAP, a nonlinear dimensionality reduction technique that preserves both local and global structure (McInnes et al., 2018). The SHAP value matrix  $\Phi$  was projected into a two-dimensional embedding  $\Psi \in \mathbb{R}^{N \times 2}$  as:

$$\Psi = \text{UMAP}(\Phi), \quad \Psi \in \mathbb{R}^{N \times 2}$$
, (5)

Each point in the two-dimensional UMAP space represents a sample characterized by its SHAP value contribution pattern and is color-coded according to its LULC category. This SHAP-UMAP visualization enables the identification of clusters exhibiting similar model behaviours associated with distinct LULC types, thereby enhancing interpretability by linking model responses to recognized landscape classes. Together, permutation importance, SHAP values, and UMAP visualization form a robust interpretability framework, ensuring that predictions from the PIML model reflect physically meaningful relationships.

#### 3 Results




## 3.1 Temporal changes in climate and land surface processes

As essential components of the TTOP model, the air freezing index (AFI), air thawing index (ATI), and LULC serve as key indicators of climate change and directly influence the TSP. Accordingly, we analysed the spatiotemporal trends of AFI, ATI, and land use and cover changes (LUCC) under four Shared Socioeconomic Pathways (SSPs). Figure 1 shows the projected temporal trajectories of AFI and ATI from 2015 to 2100, calculated as the arithmetic average across 14 CMIP6 climate models. Under all SSPs, AFI consistently declines while ATI increases. Specifically, average AFI is projected to decline at annual rates of –2.1 °C·d/yr under SSP126 and up to –10.0 °C·d/yr under SSP585. In contrast, ATI is projected to increase at annual rates ranging from +3.2 °C·d/yr under SSP126 to +17.4 °C·d/yr under SSP585, with the largest changes occurring in high-emission scenarios after 2040. These trends align well with the phenomenon of Arctic amplification (You et al., 2021) and suggest rising thermal stress on permafrost. However, substantial inter-model variability has been observed. For example, under SSP126, the CMCC-ESM2 model projects the largest decline in AFI (–7.0 °C·d/yr), whereas the MRI-ESM2-0 model indicates a slight increase (+1.0 °C·d/yr), highlighting divergent responses in cold-season conditions. Under SSP585, AFI decline rates would range from –14.9 °C·d/yr (IPSL-CM6A-LR)to –6.7 °C·d/yr (INM-CM5-0). Similarly, projected ATI increase rates would range from +0.2 °C·d/yr (MRI-ESM2-0) to +27.5 °C·d/yr (CanESM5). A full comparison of model-specific rates is provided in Table S6.



**Figure 1.** Trends and changes in air freezing and thawing indices (AFI and ATI) from 2015 to 2099 under four SSP scenarios (SSP126, SSP345, SSP370, and SSP585) in Northeast China. The indices represent the arithmetic averages computed from site-level downscaled data at 225 meteorological stations, using the delta downscale method applied across 14 CMIP6 models. Trends were estimated using the Sen's slope method.

To further examine the spatial dynamics of freeze-thaw responses, we have conducted a case analysis under the low-emission scenario SSP126 (Fig. 2). The results reveal pronounced amplification effects of the AFI along both latitudinal and elevational gradients. Specifically, annual AFI declines are projected to be more pronounced in higher-latitude and higher-elevation regions, consistent with the effects of Arctic amplification and elevation-dependent warming. Between the 2020s and 2100s, AFI reductions are projected to exceed –250 °C·yr in northern mountainous areas, whereas relatively moderate declines are expected in southern lowland zones.

In contrast, ATI is projected to exhibit an inverse gradient. While increases in ATI are expected across all zones, the rate of amplification is lower at higher latitudes and elevations. The most substantial increases are anticipated in low- to midelevation areas, likely driven by urban expansion and changes in land use. This spatial asymmetry in ground freeze—thaw responses suggests that reductions in freezing intensity are primarily driven by climate warming, particularly in colder regions, whereas increases in thawing intensity are more constrained and potentially moderated by local surface conditions and anthropogenic disturbances.



**Figure 2.** Latitudinal and elevational gradients of changes in air freezing and thawing index changes in Northeast China from the 2020s to the 2100s under SSP126. Notes: Insets (a) and (b) show latitudinal and elevational changes in the air freezing index (ΔAFI), respectively, and; insets (c) and (d) depict latitudinal and elevational changes in the air thawing index (ΔATI) between across four future periods (2040s, 2060s, 2080s, 2100s), relative to the 2020s baseline.

Similarly, significant LUCCs are projected (Fig. 3a and 3b). Between 2020 and 2100, cropland area is expected to decline slightly from 454,148 to 432,359 km², while impervious surfaces are projected to expand substantially from 43,763 to 95,973 km². Forest and grassland areas are expected to contract steadily, especially in the south-central transitional zones. As shown in Fig. 3b, land use patterns may exhibit distinct temporal transitions over the 21st century. Forest remains the most stable cover type, with minimal change across all time periods. In contrast, cropland is projected to undergo frequent transitions, although its total area is expected to remain relatively stable. This stability is primarily driven by substantial inflows from grassland conversion, with more than 5,000 km² of grassland projected to be converted to cropland between 2020 and 2040. Impervious surfaces (urban land) are projected to experience the most dramatic expansion, doubling in area from 43,763 km² in 2020 to 95,973 km² by 2100, primarily through conversions from cropland and grassland. These transitions reflect intensifying anthropogenic disturbances, particularly in south-central parts of Northeast China. Such land transformations are likely to accelerate ecosystem fragmentation and reduce surface thermal buffering capacity, ultimately increasing permafrost vulnerability. Together, these results indicate a dual pressure mechanism: rising potential ground thawing due to climate warming, and localized permafrost disturbance driven by anthropogenic land transformation. Both factors have been incorporated as dynamic inputs in our integrated PIML-based permafrost modelling framework.

330

Figure 3. Projected land use and land cover (LULC) changes in Northeast China from 2020 to 2100 and associated area transitions. Notes:

(a) Spatial distribution of dominant LULC types at decadal intervals, and; (b) Sankey-style visualization of areal changes among seven LULC classes.

## 3.2 Assessment of model performance and MAGT estimation accuracy

To evaluate the capability of different ML algorithms in simulating TSP, we have compared model-predicted MAGTs with observed values using six ML methods: decision tree (DT), multilayer perceptron (MLP), CatBoost, random forest (RF), support vector machine (SVM), and extreme gradient boosting (XGB). The validation results are summarized in Fig. 4. All models have successfully captured the overall trends of MAGT variations. However, their predictive accuracies vary notably. Among them, CatBoost and MLP exhibit the best performance, achieving the lowest mean squared error (MSE) (1.21 and  $1.28 \,^{\circ}$ C) and highest R² values (0.89 and 0.88), respectively (Fig. 4b and 4c). The RF and XGB models also demonstrate strong performance (R² = 0.85), albeit with slightly higher RMSE values of 1.25 and 1.27 °C, respectively. The DT model show moderate performance (R² = 0.74), whereas SVM model yields the weakest fit (R² = 0.69; RMSE = 1.81 °C), likely due to its limited ability to generalize nonlinear and high-dimensional relationships under complex variable interactions.

These results indicate that ensemble learning algorithms, particularly CatBoost and XGB, consistently outperform single-tree or kernel-based methods in modelling MAGT across ecologically heterogeneous landscapes. CatBoost, in particular, demonstrates strong robustness due to its capacity to handle missing data and categorical variables while mitigating overfitting, making it a reliable core model within the proposed PIML framework. The high-performing models are subsequently employed to simulate future MAGT patterns and permafrost extent under projected scenarios.

**Figure 4.** Validation of mean annual ground temperature (MAGT) predictions using six machine learning models. Notes: Panels (a–f) compare model-predicted MAGT values with observed ones for: (a) decision tree (DT), (b) multilayer perceptron (MLP), (c) CatBoost, (d) random forest (RF), (e) support vector machine (SVM), and; (f) extreme gradient boosting (XGB). Each panel includes an 1:1 reference line (dashed), a linear regression fit (red), and a 95% confidence band (shaded). Performance metrics (MSE, RMSE, and R²) are shown in each plot.

# 3.3 Projected changes in MAGT and permafrost extent

Simulations based on the PIML framework indicate a substantial and accelerating degradation of permafrost across Northeast China throughout the 21st century. This process is closely tied to rising ground temperatures and a progressive reorganization and northward shifting of permafrost zones, with the severity of change varying across emission scenarios. As shown in Fig. 5, MAGT exhibits a clear rising trend across all SSPs, with the most pronounced increases occurring under

SSP370 and SSP585. By the end of the 21st century, most lowland and mid-elevation regions are projected to exceed the 0 °C threshold of MAGT, signalling widespread thermal destabilization of permafrost. Even under the low-emission scenario of SSP126, permafrost degradation is evident, with the SLLP and permafrost zones gradually shifting northward.

Figure 5. Projected mean annual ground temperatures (MAGTs) across Northeast China under four SSPs scenarios based multilayer perceptron model.

Correspondingly, the areal extent and spatial continuity of permafrost zones undergo notable reductions (Fig. 6).

Discontinuous permafrost is replaced by sporadic or isolated patches over time. Under high-emission pathways (SSP370 and SSP585), discontinuous near-surface permafrost is projected to nearly disappear by the 2080s. These transformations begin on the northern Song-Nen river Plain, eventually encroaching upon the central Da and Xiao Xing'anling mountains.

Figure 6. Projected changes in permafrost extent in Northeast China under four SSPs scenarios based on multilayer perceptron model.

- Across all future scenarios (SSP126–SSP585), the MLP-TTOP model projects a substantial decline in total permafrost extent by the end of the 21st century. As shown in Fig. 7, under the low-emission SSP126 scenario, the total permafrost area is expected to shrink from 2.51×10<sup>5</sup> km<sup>2</sup> in the 2040s to 2.16×10<sup>5</sup> km<sup>2</sup> in the 2100s, i.e., a reduction of ~14%. In stark contrast, under the high-emission SSP585 scenario, permafrost extent is projected to plummet from 2.39×10<sup>5</sup> km<sup>2</sup> in the 2040s to just 6,000 km<sup>2</sup> by 2100, a loss of over 97%.
- Regionally, the DXAM retain the highest absolute permafrost coverage, followed by the Hulun Buir Plateau (HBP). The most severe losses are projected to happen on the northern Songhua-Nen rivers Plain (NSNP) and in the XXAM, where permafrost is expected to nearly vanish by mid-century under SSP370 and SSP585. Notably, even under the mitigated SSP126 pathway, significant degradation of permafrost is projected near the SLLP, indicating heightened sensitivity to warming.
- These findings highlight the pronounced spatial heterogeneity of climate-induced permafrost degradation and underscore the disproportionate vulnerability of discontinuous and sporadic permafrost zones, particularly in ecologically sensitive transitional areas, to ongoing and future climate warming.

Figure 7. Temporal change of permafrost area in Northeast China and subregions under four SSPs scenarios based on multilayer perceptron model results. (Notes: Panels a to e: Temporal change of permafrost area in (a) total Northeast China; (b) Xiao Xing'anling Mountains; (c) Da Xing'anling Mountains; (d) northern Songhua-Nen rivers Plain, and; (e) Hulun Buir Plateau)

# 3.4 Interpretation of key predictors: SHAP and model-based importance

To identify the key drivers of permafrost extent and degradation, predictor importance was evaluated using both modelderived rankings and SHAP values. Fig. 8 summarizes the relative importance of 15 environmental variables across six ML models.

Mean annual land surface temperature (MALST) consistently ranks as the most influential predictor across all models, highlighting its dominant role in governing ground thermal regimes. Its high average importance and low inter-model variability, particularly in ensemble models, such as RF, CatBoost and MLP, highlight its robustness across algorithmic approaches. LULC also emerges as a key predictor, ranking among the top two predictors in three out of six models. This reflects the strong influence of surface cover and anthropogenic modifications on near-surface energy exchange and permafrost stability.

Other cryo-climatic variables, such as mean annual snow cover duration (MASCD), latitude and elevation, show moderate-to-high importance. These likely capture the combined effects of snow insulation and latitudinal/elevational climate gradients. In contrast, topographic and edaphic variables, such as slope angle, slope aspect, soil organic matter contents (SOC), bulk density (BD), and sand and clay contents, generally rank lower, though they may modulate soil thermal properties and hydrological processes as secondary controls.

**Figure 8.** Model-derived feature importance rankings across six machine learning algorithms. (Acronyms: MALST represents mean annual land surface temperature; LULC, land use and land cover; MASCD, mean annual snow cover duration; SOC, soil organic carbon; BD, bulk density; MANDVI, multilear average normalized difference vegetation index; DT, decision tree; MLP, multileayer perceptron; RF, random forest; SVR, support vector regression, and; XGB, XGBoost)

Interestingly, while elevation is not consistently ranked highly, SHAP dependence plots reveal nonlinear relationships, particularly under ~1000 m a. s. l., likely reflecting microclimatic or terrain-induced variability (Fig. 9). Among edaphic predictors, SOC, BD, and clay content exhibit moderate effects with unimodal or threshold-type SHAP patterns, suggesting their roles in modulating soil thermal conductivity and moisture retentions.

Topographic predictors, such as slope angles, slope aspect, and normalized difference vegetation index (NDVI), show limited importance, though asymmetries in SHAP values indicate minor contributions. The UMAP projection (Fig. 9n) further supports the distinctiveness of LULC categories in predictor space. Forest and cropland form well-separated, compact clusters, linked respectively to lower MALST, higher SOC, and longer SCD in forests, versus flatter terrain and altered albedo in croplands. Urban (impervious) surfaces form in a narrow, isolated cluster, reflecting starkly modified surface and soil conditions. Partial overlap between grassland and barren land may indicate transitional ecotones or similar soil-climate features in degraded landscapes.

Figure 9. SHAP-based interpretation of key predictors and LULC clustering. (Notes: Panels a to m show SHAP dependence plots for major predictors. Panel n presents a Uniform Manifold Approximation and Projection (UMAP) visualization of land use and land cover (LULC) categories in the multidimensional predictor space) Acronyms: SHAP stands for SHapley Additive exPlanations; MALST, mean annual land surface temperature; LULC, land use and land cover; MASCD, mean annual snow cover duration; SOC, soil organic carbon; BD, bulk density, and; MANDVI, multiyear average normalized difference vegetation index.

Together, the model-derived and SHAP-based analyses reveal a hierarchical structure of environmental controls on permafrost distribution. Thermal variables and land cover dominate, followed by snow-related, geographic, and edaphic

factors. These findings provide physically plausible insights into permafrost dynamics and validate the interpretability of the ML framework.

## 4 Discussion




## 4.1 LULC dynamics in permafrost modelling

This study has demonstrated the significance of LUCC in modeling permafrost dynamics. Traditional permafrost models often assume static LULC and minimal anthropogenic influences in permafrost regions. For instance, many simulations over the Qinghai-Tibet Plateau presume relatively stable LULC over recent decades (Hu et al., 2022; Zhang et al., 2022; Pan et al., 2023; Li et al., 2024a), thereby underestimating the scale and impact of anthropogenic disturbances.

Such assumptions overlook the critical role of surface disturbances (e.g., deforestation, wetland drainage, infrastructure development, or wildfires) in reshaping the land–atmosphere energy exchange. These processes alter surface roughness, albedo, vegetation cover, and soil insulation, leading to profound changes in ground thermal regimes (Carpino et al., 2021; Jin et al., 2024; Li et al., 2024b, 2024c). Ground-based observational studies have consistently shown that variability in vegetation and ground covers significantly modify net radiation, sensible and latent heat fluxes, and soil moisture, ultimately influencing the TSP (Chang et al., 2015; Fisher et al., 2016; Fedorov et al., 2019). Neglecting such processes compromises the accuracy and generalizability of permafrost models, especially in regions undergoing ecological transformation or anthropogenic disturbances.

To address these limitations, our modelling framework incorporates time-varying LULC information, enabling the simulation to account for both spatial and temporal landscape dynamics. This approach closes a critical gap in previous studies that focus predominantly on climatic drivers while neglecting direct human impacts. Our results show that areas experiencing substantial LUCC (e.g., the conversion from forest to grassland or infrastructure expansion) exhibit markedly different TSP compared to undisturbed areas. These differences are physically consistent, with the altered surface energy balance and soil thermal properties associated with LUCC.

The role of LUCC in permafrost degradation is further supported by previous work. For instance, Wang et al. (2022) showed that deforestation, agricultural development, and urbanization substantially accelerate the degradation of Xing'an permafrost (XAP) by increasing soil thermal conductivity and reducing vegetation-mediated insulation. In contrast, intact forests and wetlands act as thermal buffer through thick organic layers and high evapotranspiration, mitigating ground warming (Jin et al., 2008, 2025).

By incorporating LULC trajectories into a PIML model, we capture both climatic and anthropogenic influences on permafrost evolution, leading to a more comprehensive and realist assessment. This emphasizes the urgent need to consider land surface dynamics, not just air warming, as critical determinants of permafrost stability in a changing environment.







# 4.2 Nonlinear permafrost degradation trajectories response to climate warming

Our projections reveal a nonlinear trajectory of permafrost degradation over the 21st century. Although ground thawing persists throughout the period from 2040 to 2080, the rate of degradation markedly accelerates after 2080, particularly under high-emission scenarios. This late-century acceleration suggests that critical thermal thresholds may be crossed beyond 2080, triggering nonlinear and potentially abrupt degradation of the Xing'an permafrost.

While this study does not explicitly analyse climate anomalies, prior observations from Northeast China provide important context, reporting the localized ground cooling or delayed warming during recent decades. For instance, Chang et al. (2022) reported localized ground cooling at shallow depths and a thinned active layer in the Yituli'he region during 2009–2020, despite the broader trend of regional climate warming. These anomalous conditions were attributed to stable mean positive air temperatures, reduced snow cover, and decreased anthropogenic pressure due to local population decline, possibly followed by declining anthropogenic influences and recovering vegetation. Similarly, He et al. (2021) documented ground temperature decreases in the Nanwenghe Wetland Reserve, lined to elevated water tables and the insulating effects of wetland soil. Such findings underscore that, in some areas, even under a broader warming trend, regional climatic and ecological feedbacks (e.g., snow dynamics, vegetation succession, and surface moisture conditions), can temporarily buffer permafrost against climate warming.

However, our results suggest that while these feedbacks may temporarily moderate thaw in certain subregions, they are insufficient to halt or prevent long-term degradation. As air temperatures continue to rise, especially in the latter half of the century, these buffering mechanisms are progressively overwhelmed. This is particularly evident in regions near the southern limit of latitudinal permafrost (SLLP), such as Northeast China, where permafrost exists under marginal thermal conditions and is highly sensitive to external forcing.

The spatiotemporal heterogeneity observed in permafrost degradation patterns reflects the complex, nonlinear interplay between climate drivers and land-surface processes (Jin et al., 2025). While some areas may exhibit delayed or attenuated responses due to ecological insulation or topographic shading, these effects are limited in both magnitude and temporal persistence. Our model results clearly show that, under continued warming, even regions with transient stability are likely to experience rapid thaw once thermal thresholds are exceeded.

This trajectory highlights the need for permafrost models to account for both short-term variability and long-term trends. Apparent deceleration or temporary reversal of degradation, whether due to vegetation cover, snowpack fluctuations, or hydrological changes, should not be misconstrued as evidence of permafrost resilience. Rather, such signals reflect the thermal inertia of geocryological system prior to abrupt transitions. The sharp increase in projected degradation after 2080, particularly under high-emission scenarios (SSP370 and SSP585), serves as a clear warning: without significant mitigation of carbon emissions, the preservation of XAP at shallow depths (







# 4.3 Spatial heterogeneity and resilience of XAP

As shown in Fig. 7, permafrost in Northeast China exhibits pronounced spatial heterogeneity in its response to climate warming, particularly among the subregions of Da and Xiao Xing'anling mountains, northern Songhua-Nen Rivers Plain, and Hulun Buir High-Plain. Among these, the densely forested Da Xing'anling Mountains demonstrate relatively greater resilient, preserving permafrost longer and exhibiting slower degradation rates compared to surrounding regions. Several factors are likely to contribute to this resilience. The Da Xing'anling Mountains is characterized by dense boreal forest cover, primarily larch and other mixed boreal deciduous species, and complex topography with varied slope aspects and elevations. These ecological and geomorphic features act as natural thermal buffers. Forest canopies reduce incident solar radiation during summer, while thick organic soil layers (e.g., moss and peat) dampen seasonal temperature fluctuations and insulate the permafrost (Jorgenson et al., 2010; Guo et al., 2018). Field studies on the southern slopes of the Da Xing'anling Mountains show that undisturbed larch forests maintain relatively stable ground temperatures, in contrast to adjacent open areas that experience significant ground warming. This thermal buffering arises from a combination of surface shading, reduced radiation input, and insulation from organic-rich soils, producing a combination of surface and thermal offsets of up to 3-4°C (Chang et al., 2015; He et al., 2021). Additionally, the region's complex terrain fosters diverse microclimates: north-facing slopes and valley bottoms receive less solar radiation and tend to drain cold air, resulting in locally lower ground temperatures than surrounding lowlands or upper slopes (Jin et al., 2008, 2024; Huang et al., 2025). This persistence of XAP in this region is shaped by interacting ecological, topographic, and hydrogeological processes, garnering growing scientific interest. Numerous studies have shown that such interactions can sustain permafrost under mean annual air temperatures exceeding +2 °C (Jorgenson et al., 2010). This class of permafrost, often referred to as ecosystemdominated (i.e. -driven, -modified or -protected) permafrost, is especially prominent in the Xing'an-Baikal region, where dense forest canopies and organic-rich soils shield the subsurface from warming (Jin et al., 2007, 2025; Zhang et al., 2024c). Microtopography also plays a significant role in shaping spatial thaw dynamics. For instance, field measurements in boreal Alaska revealed that elevated, forested hummocks experienced active layer thickening of only ~8 cm/year, while nearby lowlying, sparsely vegetated areas showed rates as high as ~44 cm/year (Eklof et al., 2024). These differences highlight the influence of canopy shading and surface roughness in modulating ground heat flux. Hydrogeological conditions further affect thermal stability: long-term drying and surface drainage can increase ground albedo and reduce soil thermal conductivity, thereby cooling the ground (Göckede et al., 2019). Conversely, poor drainage or thick insulating snowpack can

Comparative studies across boreal and subarctic regions, such as Siberia, Canada, and Alaska, consistently demonstrate that vegetation, topography, and moisture regimes jointly determine permafrost vulnerability under climate change. In our study, the relative resilience of XAP reflects the synergistic effects of several of dense forests, insulating organic soils, and topographically driven microclimates. By contrast, more rapid degradation observed in the Xiao Xing'anling Mountains, northern Song-Nen river Plain, and Hulun Buir Plateau subregions likely stems from less favourable conditions: widespread

enhance heat penetration and promote thaw, even in relatively cold settings (Göckede et al., 2019).




grasslands and agriculture (with lower insulation capacity), more uniform terrain, and greater exposure to solar radiation on south-facing slopes.

These findings underscore heterogeneity in permafrost vulnerability to climate warming, even under similar climate forcing, but the vulnerability is strongly modulated by local surface characteristics. This aligns with prior research emphasizing the importance of vegetation cover, snow dynamics, and organic layers in controlling climate—permafrost interactions (Jin et al., 2024). For reliable projections, especially in the marginal zone of isolated patches of permafrost near the SLLP, it is crucial

2024). For reliable projections, especially in the marginal zone of isolated patches of permafrost near the SLLP, it is crucial that models incorporate spatial heterogeneity in land cover, terrain and soil properties.

By integrating these factors, the proposed PIML framework captures how protective ecosystems favourable microtopography can substantially delay permafrost thaw relative to more exposed landscapes. Overall, this highlights the need to embed local-scale variability into regional and global permafrost models, reducing the risk of overgeneralizing the timing and magnitude of thaw under a warming climate.

### 4.4 Further prospects and inadequacies

Many widely used permafrost models are limited by shallow simulation depths and simplified physics. These models focus on near-surface thermal conditions and may not capture permafrost extending tens of meters deep (Alfaro Sánchez et al., 2024). In Northeast China, for example, permafrost can exceed 50–100 m in thickness. Because simple models do not account for these residual permafrost layers at depths, they might overestimate the rate of thaw and exaggerate how soon permafrost will completely thaw. Actually, even after shallow layers thaw, warmer and ice-rich permafrost at depth can persist for decades, delaying complete thaw beyond what shallow models predict (Peng et al., 2023). This highlights the inadequacy of models that only simulate the upper soil and underscores that permafrost degradation is a 3-dimensional problem.

Permafrost thaw is governed by 3-dimensional processes such as lateral heat advection, moisture transport, and talik development, yet these critical physical mechanisms are often absent in traditional one-dimensional models (Sun et al., 2019). The formation and lateral expansion of taliks illustrate this complexity. Taliks can grow not just from the surface downwards, but also laterally (for instance, under thermokarst lakes or along subsurface water channels), making thaw progression highly heterogeneous (Sun et al., 2022). Lateral heat transfer through groundwater and convective flows can significantly accelerate thaw and these processes are not considered in many models (Vasheghani Farahani et al., 2021). Simulations have demonstrated that including lateral and vertical advection of heat (e.g. from infiltrating rain or snowmelt) in modelling research raises ground temperatures and advances the thaw front (Shook et al., 2024). In essence, preferential flow of water through soil macropores or along taliks delivers heat deep into the ground within a short time, greatly speeding up permafrost degradation that purely conductive models would underestimate (Walvoord and Kurylyk 2016).

In addition, the classification of permafrost presents significant challenges, particularly in regions such as the XAP zones. The conventional scheme classifies permafrost by areal continuity of permafrost (e.g. continuous (>90% area), discontinuous (50–90%), sporadic (10–50%), and isolated patches or patchy (






categories is challenging. Field surveys often reveal that mapped discontinuous or sporadic permafrost zones contain a patchwork of frozen and unfrozen ground that is difficult to quantify by aerial percentage alone (Jin et al., 2025). In XAP zones, permafrost is sustained by local boreal forest and wetland conditions and tends to be warm (near 0 °C), thin, and in a delicate thermal balance (Zhang et al., 2024c). Studies suggest a symbiotic relationship between permafrost and the overlying ecosystem that grants it a degree of ecological resilience to climate warming (Jin et al., 2008). For instance, insulating organic matter and shading effects can maintain or cool permafrost despite rising air temperatures, meaning that XAP may still survive in locations where climate-based models would predict degradation (Jin et al., 2024). Traditional area continuity-based classifications do not fully capture such resilience or the unique hydrothermal properties of XAP. Near the SLLP, finding a reliable classification scheme that reflects not just climate and area fraction but also ecological and geotechnical stability and in areal density of permafrost or in 3-dimensional distribution of dynamically changing permafrost is an open challenge (Jin et al., 2025), especially under extreme hydroclimate (e.g., heat waves, storm or melt induced floods), disruptions from natural (e.g., wildfires) or anthropogenic (e.g., engineering, land cultivation, urbanization) activities. Any new scheme must serve practical needs in ecosystem conservation, engineering stability, and hydrology, which demands integrating field observations with nuanced criteria beyond simple percentage cover.

## **5** Conclusions

To address the modelling bias introduced by assuming static LULC in permafrost prediction, this study develops a hybrid modelling framework that integrates the TTOP model with dynamic LULC simulations from the PLUS model. Meanwhile, ML models are imbedded within the TTOP structure to improve predictive performance while maintaining physical interpretability. This PIML approach offers a novel pathway for simulating the spatiotemporal evolution of permafrost under changing environmental conditions. The main findings are summarized as follows:

# 1) Model performance and predictive accuracy

Among the six ML models evaluated, the MLP and CatBoost perform exhibit the highest predictive accuracy for MAGTs. MLP achieves RMSE and MAE values of 1.13 and 1.28 °C, respectively, while CatBoost attains 1.10 and 1.21 °C. Compared to traditional physically based models, the PIML framework yields superior performance, particularly in areas near the SLLP, where the ground thermal regime is governed by complex, coupled climatic, ecological, and anthropogenic drivers. This underscores the ability of PIML to capture nonlinear interactions in transitional permafrost zones.

# 2) Importance of LUCC

LUCC, often neglected in earlier permafrost studies, emerges as a key driver of permafrost thermal dynamics. Model-based feature importance analyses highlight that MAGST, LULC, and SCD are dominant predictors of MAGT and TSP. Future land surface alterations due to urban expansion, deforestation, or other anthropogenic and natural LUCC processes will significantly affect land—atmosphere energy exchanges and ground hydrothermal dynamics. These impacts must be explicitly incorporated into next-generation permafrost models to improve their predictive fidelity under realistic landscape dynamics.

https://doi.org/10.5194/egusphere-2025-4544 Preprint. Discussion started: 17 November 2025

© Author(s) 2025. CC BY 4.0 License.





# 575 3) Future permafrost trajectories under emission scenarios

Model projections indicate strong scenario-dependent divergence in permafrost degradation. Under the low-emission SSP126 pathway, the near-surface permafrost area in Northeast China is projected to decline by □14% by the end of the 21st century. In contrast, under the high-emission SSP585 scenario, up to 97% of the current extent of near-surface permafrost is expected to be lost. Regional sensitivity also varies considerably. The northern Songhua-Nen Rivers Plain and the Xiao Xing'anling Mountains are projected to experience near-complete permafrost loss, whereas the Da Xing'anling Mountains and the Hulun Buir High-Plain demonstrate higher climatic resilience and slower thaw rates of the XAP.

# 4) Future research prospects and challenges

Despite the advances achieved in this study, considerable uncertainties remain in permafrost modeling. Current models often neglect deep residual permafrost layers and 3-dimensional hydrothermal processes such as lateral heat advection, preferential flow, and talik development, which can significantly alter thaw trajectories. Addressing these limitations requires integrating deeper subsurface processes and hydrological dynamics into next-generation models. Moreover, conventional permafrost classifications based solely on areal continuity fail to capture the ecological resilience and spatial heterogeneity of warm, thin permafrost in the XAP zone. Developing refined classification schemes that incorporate ecological, hydrothermal, and geotechnical stability criteria will be critical for improving predictions and supporting ecosystem conservation, engineering safety, and hydrological management.

#### Data availability

The air temperature datasets used in this study were obtained from the National Meteorological Information Centre of China (NMICC) under the Climatic Data Centre at the China Meteorological Administration (https://data.cma.cn/). The CMIP6 climate projection datasets were sourced from the Earth System Grid Federation (ESGF) data repository (https://esgf-node.llnl.gov/projects/cmip6/). Detailed sources of the driving variables used in the PLUS and PIML models are listed in Table S2 and Table S5 in the Supplementary Materials.

## Acknowledgements

This study was financially supported by the National Natural Science Foundation of China (Grant No. 42271130), National Key R&D Program of China (Grant Nos. 2024YFF0809102 and 2022FY100703), Heilongjiang Touyan Innovation Team Program (Forest Carbon Sink Assessment and Carbon Sequestration Management Innovation Team) and Program of China Scholarship Council (Grant No. 202406600030).

# **Competing interests**

The authors declare no competing interests

### **Author contributions**

S. H. designed the study and wrote the paper. H. J. and X. K. revised the paper and acquired project funds. X. Y., X. J., S. L., and L. Y. contributed to the discussion of the study and improved the paper. Y. Z. and H. W. collected and curated the data. K. G. provided the validation data. X. L., R. H., and L. L. contributed to software development and visualization. G. C. reviewed and edited the manuscript. All authors read and approved the final manuscript.

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
