# Peer review of "A physics-informed machine learning (PIML) framework for projecting 21st-century permafrost extent in Northeast China"

_EGUsphere, 2025_

## Community Comment (CC2)

This study develops a PIML framework that integrates physically based modeling with machine learning and incorporates dynamic land-use/land-cover changes to simulate and project permafrost evolution in Northeast China. The methodology demonstrates a certain degree of innovation. Since the study area is located near the SLLP in East Asia, its permafrost characteristics are regionally representative, and the findings provide valuable regional applicability and scientific insight. The manuscript is overall well written, but minor improvements can be made regarding clarity of expression as well as figure and text descriptions. The specific comments are as follows:

1. The abstract predominantly provides qualitative descriptions. It is recommended to include more quantitative results to enhance informativeness.

2. In Section 3.2, it is suggested to add comparisons with existing permafrost maps developed for the same region.

3. In Figure 7, please indicate the spatial extents corresponding to the Da Xing'anling Mountains, Xiao Xing'anling Mountains, the northern Song-Nen rivers Plain, and the Hulun Buir Plateau.

4. Lines 327–328 and 564–565 contain inaccurate wording, as the predictive accuracies of MLP and CatBoost differ depending on the metric used; thus, it is inappropriate to state that both models simultaneously exhibit the best performance.

5. In Lines 564–565, MAE is mentioned without prior reference, which seems to be a typographical error where MSE was mistakenly written as MAE.

6. The unit of MSE in the manuscript should be $°C^2$ instead of $°C$.

---

## Author Comment (AC1)

**Comment**:

I believe the author has done an excellent job, but I think it would be preferable to compare such predictions with permafrost mapping or field survey results from the observed time period.

**Response**:

Dear Dr. Li:

Thank you very much for your constructive suggestion. We fully agree that comparing our simulation results with existing permafrost maps and field-based evidence is essential for strengthening the reliability of model outputs. In response to your comment, we have added a comprehensive comparison between our simulated permafrost distribution during 2001–2020 and two recently published Northern Hemisphere permafrost maps (Ran et al., 2022; Obu et al., 2019). The newly added content is presented in the revised manuscript (Lines 343–364) and the comparison is illustrated in the newly added Figure 5. Revision as below:

**L343-364**:

In addition, we compared the permafrost distribution simulated by the MLP model in this study during 2001–2020 with the recently published Northern Hemisphere permafrost maps (as shown in Fig. 5). Across the three permafrost maps, we observed a consistent representation of the widespread permafrost distribution in the Da Xing'anling Mountains, with the SLLP located approximately in the Arxan mountains. However, notable discrepancies occur among studies for the permafrost distribution in the Xiao Xing'anling Mountains, the Hulunbuir Plateau, and the southern mountainous regions (Huanggangliang Mountains and Changbai Mountains). For the Xiao Xing'anling region, our results are more consistent with those of Ran et al. (2022), but differ significantly from Obu et al. (2019). According to Huang et al. (2025), the SLLP in the Xiao Xing'anling mountains is located approximately between Heihe and Bei'an, which agrees well with our simulation. For the Hulunbuir Plateau, our estimation lies between the results of Ran et al. (2022) and Obu et al. (2019). However, due to the limited availability of field observations in this area, further verification is required. Regarding SLLP characteristics, the simulated permafrost distribution near the southern boundary in this study appears more scattered, reflecting the presence of isolated permafrost patches near the SLLP. This pattern is consistent with the actual conditions. With respect to the permafrost in the southern mountainous regions of Northeast China, our results and those of Ran et al. (2022) and Obu et al. (2019) all indicate the presence of permafrost. However, Obu et al. suggest a more extensive permafrost area in the Huanggangliang mountains, whereas both our study and Ran et al. (2022) show a more sporadic distribution. Based on the synthesis by Jin et al. (2025) and field surveys, permafrost in the southern mountainous regions of Northeast China may indeed exist but is difficult to detect; its occurrence is likely controlled by local factors. These findings further support the results of this study.

[Figure]

**Figure 5.** Comparison of the permafrost zone between the results of this study, Ran et al., 2022 and Obu et al., 2019. Notes: (a) result of this paper based on multilayer perceptron (MLP) model during 2001-2020, (b) result of Ran et al., 2022 during 2000-2016 and (c) result of Obu et al., 2019 during 2000-2016.

---

## Author Comment (AC2)

**Comments**:

This study develops a PIML framework that integrates physically based modeling with machine learning and incorporates dynamic land-use/land-cover changes to simulate and project permafrost evolution in Northeast China. The methodology demonstrates a certain degree of innovation. Since the study area is located near the SLLP in East Asia, its permafrost characteristics are regionally representative, and the findings provide valuable regional applicability and scientific insight. The manuscript is overall well written, but minor improvements can be made regarding clarity of expression as well as figure and text descriptions. The specific comments are as follows:

**Response:**

Dear Dr. Hu,

We sincerely appreciate your positive and encouraging feedback on our manuscript. Your thoughtful evaluation and recognition of our work are truly motivating.

We would also like to thank you for the time and effort you devoted to reviewing our paper, and for the constructive suggestions you provided. These comments have been very helpful in further improving the clarity and rigor of our work.

1. The abstract predominantly provides qualitative descriptions. It is recommended to include more quantitative results to enhance informativeness.

**Response:**

Thank you for the valuable comment. We agree that the original abstract focused mainly on qualitative descriptions. In the revised version, we have incorporated quantitative projections of permafrost extent under different emission scenarios (SSP1-2.6 and SSP5-8.5), including absolute areas and percentage changes by the end of the 21st century, to improve clarity and informativeness. We made a modification as below (text in red indicates the revised content):

**Abstract.** The degradation of marginal permafrost is a sensitive indicator of climate change, with far-reaching implications on regional ecosystems, hydrology, and infrastructure. Located near the southern limit of latitudinal permafrost (SLLP) in Eastern Asia, Northeast China has experienced pronounced permafrost retreat and persistent ground warming in recent decades. This study develops a physics-informed machine learning (PIML) framework that integrates the Temperature at the Top of Permafrost (TTOP) model, land-use/land-cover (LULC) transitions, and CMIP6 climate projections to improve the understanding and prediction of regional permafrost dynamics. **Quantitative projections show that permafrost extent will decline by ~14% under SSP1-2.6 (from $2.51 \times 10^5$ km$^2$ in the 2040s to $2.16 \times 10^5$ km$^2$ in the 2100s), while under SSP5-8.5 it will experience a catastrophic loss exceeding 97% (from $2.39 \times 10^5$ km$^2$ to only ~6,000 km$^2$ by 2100).** This retreat is primarily driven by a sharp decrease in the air freezing index (AFI), especially in high-latitude and high-elevation zones, and is further exacerbated by urban expansion and deforestation. Mountainous forested areas (e.g., Da Xing'anling Mountains) exhibit comparatively greater resilience due to dense vegetation and topographic buffering of surface energy fluxes. Feature attribution analysis identifies surface temperature, snow cover duration, and vegetation as dominant controls on permafrost stability, while UMAP clustering reveals distinct degradation trajectories among different land cover categories. This study highlights the complex interplay of

climatic and anthropogenic drivers in permafrost evolution and demonstrates the value of integrating physical modelling with machine learning for informing ecological conservation and infrastructure risk management in cold-region environments.

2. In Section 3.2, it is suggested to add comparisons with existing permafrost maps developed for the same region.

**Response:**

Thanks for your meticulous review. We agreed to your advice and made a revision as below:

**L343-364**:

In addition, we compared the permafrost distribution simulated by the MLP model in this study during 2001–2020 with the recently published Northern Hemisphere permafrost maps (as shown in Fig. 5). Across the three permafrost maps, we observed a consistent representation of the widespread permafrost distribution in the Da Xing'anling Mountains, with the SLLP located approximately in the Arxan mountains. However, notable discrepancies occur among studies for the permafrost distribution in the Xiao Xing'anling Mountains, the Hulun Buir Plateau, and the southern mountainous regions (Huanggangliang and Changbai mountains). For the Xiao Xing'anling region, our results are more consistent with those of Ran et al. (2022), but differ significantly from Obu et al. (2019). According to Huang et al. (2025), the SLLP in the Xiao Xing'anling Mountains is located approximately between Hei'he and Bei'an, which agrees well with our simulation. For the Hulun Buir Plateau, our estimation lies between the results of Ran et al. (2022) and Obu et al. (2019). However, due to the limited availability of field observation of data in this area, further verification is required. Regarding SLLP characteristics, the simulated permafrost distribution near the southern limit in this study appears more scattered, reflecting the presence of isolated permafrost patches near the SLLP. This pattern is consistent with the actual conditions. With respect to the permafrost in the southern mountainous regions of Northeast China, our results and those of Ran et al. (2022) and Obu et al. (2019) all indicate the presence of mountain permafrost. However, Obu et al. (2019) suggest a more extensive permafrost area in the Huanggangliang Mountains, whereas both our study and Ran et al. (2022) show a more sporadic distribution. Based on the synthesis by Jin et al. (2025) and field surveys, permafrost in the southern mountainous regions of Northeast China may indeed exist but is difficult to detect; its occurrence is likely controlled by local factors. These findings further support the results of this study.

[Figure]

**Figure 5.** Comparison of the permafrost zone between the results of this study, Ran et al. (2022) and Obu et al. (2019). Notes: (a) result of this paper based on multilayer perceptron (MLP) model

during 2001-2020, (b) result of Ran et al. (2022) during 2000-2016, and (c) result of Obu et al. (2019) during 2000-2016.

3. In Figure 7, please indicate the spatial extents corresponding to the Da Xing'anling Mountains, Xiao Xing'anling Mountains, the northern Song-Nen rivers Plain, and the Hulun Buir Plateau.

**Response:**
Thanks for your suggestions. We have modified Figure 7 as below:

[Figure]

Figure 7. Temporal change of permafrost area in Northeast China and subregions under four SSPs scenarios based on multilayer perceptron model results. (Notes: Panels (a) to (e): Temporal change of permafrost area in (a) total Northeast China; (b) Xiao Xing'anling Mountains; (c) Da Xing'anling Mountains; (d) northern Songhua-Nen rivers Plain, and; (e) Hulun Buir Plateau)

4. Lines 327–328 and 564–565 contain inaccurate wording, as the predictive accuracies of MLP and CatBoost differ depending on the metric used; thus, it is inappropriate to state that both models simultaneously exhibit the best performance.

**Response:**
Thank you for the comment. We agree that our original wording was inaccurate. We have revised the sentences in Lines 327–328 and 564–565 to clarify that CatBoost achieves the lowest MSE, while MLP attains the highest R² value. The revised statement now accurately reflects the distinct strengths of the two models as below:

**L327-329:**
Among them, CatBoost exhibits the best overall performance, achieving the lowest mean squared error (MSE) (1.21 °C$^2$) and the highest $R^2$ value (0.89), while MLP also performs competitively with an MSE of 1.28 °C$^2$ and an $R^2$ of 0.88 (Fig. 4b and 4c).

**L587-589:**

Among the six ML models evaluated, CatBoost exhibits the highest predictive accuracy for MAGTs, achieving RMSE and MSE values of 1.10 °C and 1.21 °C$^2$, respectively, while MLP also performs competitively with RMSE and MSE values of 1.13 °C and 1.28 °C$^2$.

5. In Lines 564–565, MAE is mentioned without prior reference, which seems to be a typographical error where MSE was mistakenly written as MAE.

**Response:**

Thank you for your careful observation. We confirm that "MAE" was mistakenly written instead of "MSE" in Lines 564–565. This typo has been corrected to "MSE" in the revised manuscript.

6. The unit of MSE in the manuscript should be °C$^2$ instead of °C.

**Response:**

Thank you for pointing this out. We have thoroughly checked the entire manuscript and replaced all incorrect MSE units (°C) with the correct ones (°C$^2$) to ensure consistency and accuracy.